# Consequences of the COVID-19 Syndemic for Nutritional Health: A Systematic Review

**DOI:** 10.3390/nu13041168

**Published:** 2021-04-01

**Authors:** Cristian Neira, Rejane Godinho, Fabio Rincón, Rodrigo Mardones, Janari Pedroso

**Affiliations:** 1Escuela de Psicología, Facultad de Ciencias Sociales y Comunicación, Universidad Santo Tomás, Concepción 4030585, Chile; mardrodrigo@gmail.com; 2Bolsista CNPq, Programa de Pós-Graduação em Teoria e Pesquisa do Comportamento, Universidade Federal do Pará, Belém 66075-110, Brazil; 3Programa de Pós-Graduação em Teoria e Pesquisa do Comportamento, Universidade Federal do Pará, Belém 66075-110, Brazil; rejane.godinho.2012@gmail.com; 4Programa de Pós-Graduação em Psicologia, Universidade Federal do Pará, Belém 66075-110, Brazil; faru1095@gmail.com (F.R.); pedrosoufpa@gmail.com (J.P.); 5Bolsista Produtividade CNPq nível 2, Programa de Pós-Graduação em Teoria e Pesquisa do Comportamento, Universidade Federal do Pará, Belém 66075-110, Brazil

**Keywords:** confinement, social distancing, social isolation, quarantine, pandemic, syndemic, SARS-CoV-2, COVID-19, eating behavior, feeding behavior

## Abstract

Confinement at home, quarantine, and social distancing are some measures adopted worldwide to prevent the spread of Severe Acute Respiratory Syndrome Coronavirus 2 (SARS-Cov-2), which has been generating an important alteration in the routines and qualities of life of people. The impact on health is still being evaluated, and consequences in the nutritional field are not entirely clear. The study objective was to evaluate the current evidence about the impact that preventive measures of physical contact restriction causes in healthy nutrition. A systematic review was carried out according to the “Preferred Reporting Items for Systematic Reviews and Meta-Analyses” PRISMA Group and Cochrane method for rapid systematic reviews. Searching was performed in six electronic databases and evaluated articles published between 2010 and 2020, including among their participants adult subjects who had been exposed to the preventive measures of physical contact restriction. Seven studies met the selection criteria and reported an overall increase in food consumption, weight, Body Mass Index (BMI), and a change in eating style. Findings suggest that healthy nutrition is affected by preventive measures to restrict physical contact as a result of the COVID-19 syndemic.

## 1. Introduction

One year after the confirmation of the first positive case of Severe Acute Respiratory Syndrome Coronavirus 2 (SARS-Cov-2) in the province of Wuhan, China, the virus that produces COVID-19 is still active around the world, having killed more than 1,500,000 people, with a progressive increase in the number of infected cases, which has already exceeded 69,000,000 and continues to generate various global consequences that have altered the rhythm and daily life of millions of people around the planet [1,2,3,4].

Following the recommendations of the World Health Organization (WHO), nations around the world have adopted different control measures to cut the line of viral transmission and/or mitigate the spread of SARS-CoV-2 [5]. Among the most important measures that have had the greatest impact on the population are the closing of borders, home confinement, quarantine, and physical distancing. Adopting these measures was necessary to reduce the spread of the COVID-19 pathogen. However, the consequences in different areas of health and life quality have not been fully clarified. The impact includes not only biological but also social aspects, and has gone beyond the pandemic classification to the point of being re-named as a syndemic (syn of synergy and demy of pandemy), given the synergistic nature between SARS-CoV-2, the different non-communicable diseases (NCDs), and other problems of a social and economic nature [6].

In addition to modifying lifestyle, measures of home confinement, quarantine, and physical distancing are causing serious alterations in people’s health. For example, a deterioration in mental health has been seen, mainly associated with increased levels of stress, anxiety, and depression [7,8,9,10,11], which, in turn, are considered a risk factor for increasing food intake, especially less healthy foods [12,13,14,15]. Furthermore, these measures have the characteristics to restrict people’s mobility, leading to a decrease in physical activity [16]. Then, there is a risk factor for other NCDs associated with nutritional health, such as high blood pressure, diabetes, and obesity; diseases that are severely affected by the COVID-19 syndemic [17,18,19,20].

Considering the different disorders causing this syndemic and that correct nutrition is an essential element to strengthen the immune system against the negative consequences of COVID-19 and other NCDs, different recommendations have been made to adopt a healthier eating behavior, essentially focusing on limiting the intake of salt, free sugars, saturated fats, increasing fiber consumption, and maintaining good hydration, among others [21]. However, following these recommendations does not seem to be easy, especially due to the context of stress and anxiety that the general population is experiencing, where eating unhealthy but highly tasty foods seems to be the alternative to reduce the unpleasant condition generated in the COVID-19 syndemic [15].

Given that the measures adopted to prevent the spread of the COVID-19 pathogen are unprecedented and their effects on health are not fully clarified, there is a need to understand their scope, and the scientific community has conducted several studies with different results. In this way, analyzing and evaluating current evidence on the impact that preventive measures to restrict physical contact (home confinement, quarantine, and physical distancing) are causing on people’s nutritional health became the objective of this present systematic review.

## 2. Materials and Methods

### 2.1. Study Design

This systematic review was carried out following the methodological guide The Interim Guidance from the Cochrane Rapid Reviews Methods Group [22], and was accomplished under the standards established in “Preferred Reporting Items for Systematic Reviews and Meta-Analyses” (PRISMA) [23]. A review protocol was registered in the International Prospective Registry of Systematic Reviews—PROSPERO (Registry number: CRD42020197618), and can be consulted for more information on the review design.

### 2.2. Search Strategy

Searching was made in these electronic databases: PubMed, APA PsycNet (American Psychological Association), Web of Science (Science and Social Science Citation Index), EMBASE, Scopus, and Cochrane. Downloaded articles were considered empirical, original, complete, free, and published in English between January 2010 and July 2020, and considered within their participants adult subjects, without restrictions on sex, race, or educational level, belonging to cohorts of a population that has been subjected to preventive measures to restrict physical contact, such as: social isolation, social/physical distancing, home confinement, and quarantine.

For this review, studies with a methodological design considering underage participants and subjects who have not been exposed to any preventive measures of physical contact restriction (social isolation, physical/social distancing, confinement at home, and quarantine) were excluded. Studies that were a literature review (any type), meta-analysis, involved case studies, not freely accessible, and not published in the English language were also disregarded.

The controlled vocabulary browser MeSH (Medical Subject Headings) was used to establish these keywords: social distancing, physical distancing, social isolation, quarantine, pandemic, loneliness, SARS-CoV, COVID-19, eating behavior, feeding behavior, and eating habits. Furthermore, as a way to retrospectively broaden the search, the keywords epidemic, Ebola, and H1N1 were included, especially considering these last two viruses have some similar characteristics to SARS-CoV-2 in the transmission routes and in the impact generated on the population.

The searching strategy consisted of a simple, binary combination, which included all keywords. For a better understanding of that and the Boolean operators used, see Appendix A.

### 2.3. Studies Selection

Two reviewers with experience in the thematic area of eating behavior, Rodrigo Mardones and Fabio Rincón, separately made a preliminary reading of study titles and abstracts published between 2010 and 2020 in the electronic databases mentioned above that related the preventive measures of physical contact restriction with people’s eating behavior. Titles or abstracts, including relevant information to this review, were selected and retrieved for in-depth reading, which was carried out by two other reviewers (Cristian Neira and Rejane Godinho), separately and independently. The discrepancies that occurred in the selection of articles, both in the preliminary reading and in the in-depth reading, were solved by discussion with a third reviewer (J.P.) to get consensus.

### 2.4. Data Extraction

After study selection, two reviewers (Cristian Neira and Rejane Godinho) extracted data separately using a standardized electronic form. The following information was collected from the articles: title, keywords, publication year, database, journal name, area, author(s), study objectives, context, place, country, participants, research design, instruments, techniques, procedure, statistical tests, results, and conclusions. This procedure included a pilot test to train reviewers in data extraction and coding. Discrepancies in data extraction were solved by the intervention of a third reviewer to reach consensus.

### 2.5. Methodological Quality Assessment

For the quality assessment of each eligible study included in the review, an abbreviated form of the Ottawa–Newcastle (NOS) scale was applied for cross-sectional and observational studies [24]. The NOS contains a checklist made up of three criteria: selection (representativeness of individuals), comparability (determination of confusion), and results (evaluation and results analysis). According to the scoring system, studies were scored on a range of 0 to 10 points, and classified as having a low (10 and 9 points), medium (7 and 8 points), or high (<7 points) risk of bias. In general, the NOS scale shows good reliability among evaluators and test–retest [25].

## 3. Results

The searches carried out in the online databases identified 1044 publications, of which 271 were duplicates. After exclusion of duplicates and selection through abstracts and titles, 17 articles were chosen for full-text selection. Of this total, 10 did not meet the proposed inclusion criteria, leaving seven eligible articles. Figure 1 presents the complete selection process for articles included in the analysis.

### 3.1. Methodological Quality Assessment

Of the seven studies, five were classified as having a moderate risk of bias [26,27,28,29,30] and two studies a high risk of bias [31,32]. Regarding sample selection, it was representative in all studies and, in six of them, the size was justified [26,27,28,29,30,31]; however, none used a random selection. Concerning instruments, all studies used self-report measures. One study reported data about its validation [29], and the remaining six made available or described the measurement tools used [26,27,28,30,31,32]. As for comparability, six studies were able to control the main confounding factors [26,27,28,29,30,32], and five of them managed additional factors [26,27,28,29,30]. The scores for each criterion proposed by the NOS and the total score are shown in Figure 2 and Appendix B.

### 3.2. Studies Characteristics

The seven selected studies were published in 2020, and analyzed the relationship between eating behavior and preventive measures to restrict physical contact during the COVID-19 syndemic. In six studies, population samples from different countries were examined, such as: Italy, Spain, Poland, Turkey, and Australia. One study evaluated an international sample that included cohorts from Asia, Africa, Europe, and other countries. The number of participants per sample ranged between 600 and 7514 subjects. All studies reported cross-sectional data analysis corresponding to the COVID-19 syndemic context. The characteristics and main results of the studies are showed in Table 1.

### 3.3. Eating Behavior and Confinement or Social Distancing Measures

In general, all selected studies analyzed the relationship between eating behavior and the different preventive measures of physical contact using self-report instruments (online questionnaires), which provided information exclusively on this behavior during the COVID-19 syndemic emergency. It is worth mentioning that, in the last 10 years, no relevant studies have been reported on the relationship or effect of preventive measures to restrict physical contact and eating behavior associated with the H1N1 epidemic in 2009 and Ebola in 2015. All selected studies supplied at least one main outcome (Table 1) on eating behavior, organized and categorized as follows: change in food intake [26,27,28,30,32], variation in weight and BMI [27,30], and change in eating style [26,29,31].

### 3.4. Changes in Food Intake

Three studies described a relationship between changes in food intake associated with preventive measures of restricting physical contact. One of them, carried out with an international sample [26], reported that, during the blockade by COVID-19, the number of main meals, unhealthy foods, and snack consumption increased (*t* = −5.83, *p* < 0.001, d = 0.22; *t* = −3.46, *p* < 0.001, d = 0.14; *t* = −6.89, *p* < 0.001, d = 0.30, respectively) compared with the period prior to the lockdown. Another investigation, accomplished in Spain [28], mentioned that, during the confinement period, there was an increase in the level of adherence to the Mediterranean diet compared with the time before the confinement (Md = 6.0 (1–13); Md = 7.0, (1–13), *p* < 0.001). The latest study, also made in Spain, indicated that a little over 50% of the population increased the consumption of sweets, increased 30% to 40% of the consumption of meats, fruits, eggs, rice, pasta, bread, sausages and cold cuts, vegetables, and dairy products, and 33% decreased fish consumption [32].

Additionally, two studies connected changes in food intake to BMI and age. The first one, developed in Poland [30], indicated a difference in BMI in people who did not change their food intake versus those who did change it during the COVID-19 emergency (M = 22, 9 ± 4.6; M = 24.1 ± 5.1, *p* < 0.01). The other study carried out in Italy [27] pointed out that, during the COVID-19 emergency, there was a difference between the decrease in the consumption of junk food (29.8%) versus the increase of this kind of food (25.6%) (*r*^2^  = 9.560, *p* = 0.002). This difference is associated with BMI (OR = 1.025, *p* = 0.005) and age (OR = 0.979, *p* < 0.001).

### 3.5. Weight Gain and BMI

Two studies provided results that connected weight gain to BMI and other variables during the COVID-19 syndemic. The first one observed that weight gain was correlated with BMI (Rs = 0.21, *p* < 0.05) and age (Rs = 0.15, *p* < 0.05); this relationship was more accentuated in individuals who were overweight, obese, and older than 35 years [30]. The second study showed that, during the emergence of COVID-19, the perception of weight gain was positively and inversely associated with a higher consumption of food (junk food: OR = 3.122, *p* < 0.001 and healthy food: OR = 0.805, *p* = 0.002), a higher BMI (OR = 1.073, *p* < 0.001), and the modification of labor routine during the syndemic (OR = 1.250, *p* = 0.037) [27].

### 3.6. Change in Eating Style

Three studies gave results that revealed an alteration in the eating style of people linked to physical contact restriction measures. The first of them [29], using the DEBQ (Dutch Eating Behavior Questionnaire), revealed that people who remained isolated increased behavior external eating versus those who did not (Md = 26 (20–30); Md = 23 (15–28.5)). The results of the second study, carried out with an international sample [26], showed that, during confinement at home, the uncontrolled eating behavior increased compared with the period before the emergence of COVID-19 (*t* = −9, 44, *p* < 0.001, d = 0.22). Finally, a study made in Australia analyzed the eating behavior of people with an eating disorder (ED) versus people without an ED, reporting that the frequency of eating restriction behaviors (ERBs) and compulsive eating behaviors (CEBs) increased during the emergence of COVID-19 in people without an ED by 27.6% and 34.6%, respectively, compared to the period before the syndemic. It also revealed that, in this period, the frequency was even higher in people who had an ED; thus, the level of ERBs, ECBs, and eating compulsion behavior (ECB) increased 64.5%, 35.5%, and 18.9% in people with ED, respectively [31].

## 4. Discussion

In this systematic review, we found that the preventive measures to avoid physical contact, adopted by different governments of the world to control the spread of the SARS-CoV-2 virus, modified food intake and eating style, enhancing these changes in people with a higher BMI and an ED. These results become relevant when bearing in mind that SARS-CoV-2 seriously affects people who suffer from diseases related to nutritional health, such as high blood pressure, diabetes, and obesity [17,18,19,20].

Some studies included in this review reported that the frequency of food intake increased in different categories [32], as well as the number of main meals, and the consumption of snacks and unhealthy foods [26]. Previous studies have connected increased food consumption with depressive moods [12,33] and boredom [34], as well as an increase in levels of anxiety [13] and stress [14]. These psychological changes have been enlarged during the COVID-19 syndemic [7,8,9,10,11], which suggests that the consequences on mental health produced by syndemic is spreading to people’s dietary health. This idea is in line with recent evidence showing that home confinement has negatively affected people’s lifestyles [4,35,36].

Additionally, our results showed an increase in weight and BMI during the emergence of COVID-19 [27,28], and a positive association between an enlarged intake of food and a higher BMI [30], which, in turn, was related to less healthy food choices [27]. In this sense, there is a history that people with a high BMI have of problematic eating behaviors, such as eating without hunger and in excess [34,37]. It has also been noted that people with obesity tend to eat more between meals and more frequently at night, which is detrimental to their health [38]. If we also consider the negative effects that the syndemic is causing on the population’s mental health [7,8,9,10,11], in addition to the history that links a high BMI to higher levels of stress [39,40], it is possible to understand why overweight and obese people changed their food intake during this period. At this point, we must consider that, when faced with unpleasant emotional conditions, people tend to prefer foods high in sugar and fat [15], that is, they opt for a less healthy diet, which, in turn, would influence weight gain and BMI. Another finding of this review may seem contrary to what has been proposed so far, which is the greater adherence generated to the Mediterranean diet during the COVID-19 syndemic [28], which is considered a healthy diet due to its high consumption of vegetables, legumes, fruits, cereals, and olive oil, among others [41,42]. However, the adherence that occurred during this period was mainly among people with normal weight, thus, with increasing age and BMI, less people adhered to the Mediterranean diet [27]. Additionally, it was reported that 13% of the population that adhered to this diet increased their weight [28]. The psychological impact caused by this syndrome has been associated with symptoms of food dependence [43], that is, people are eating more, regardless of whether this diet is considered healthy, since it would be a way to alleviate the psychological discomfort created during this period. In this way, excess calorie intake can lead to an increase in BMI and harm people’s health [44].

Along with changes in food intake, alterations in people’s eating style were reported during this syndemic [26,29,31]. The attitude towards food choice is influenced by different aspects [45]. For example, compulsive or uncontrolled eating during the COVID-19 emergency has been associated with people previously stigmatized for their weight [46]. Another explanation could be that this behavior is a response to emotional states related to anxiety produced by home confinement [47]. Similarly, developing external eating behavior during isolation can be an external motivational response to stress, since this behavior occurs when individuals under stress are in the same environment with food, being affected by its smell and appearance [48,49]. Thus, it can be thought that the confinement at home and easy access to food has created a favorable environment for people eating more, especially those who previously were stigmatized for their body weight.

Distinct from an uncontrolled eating style or external eating behavior, our findings also reported a higher level of restrictive eating behaviors, with the characteristics of cognitive efforts to restrict food intake. This type of behavior was recently associated with stress and anxiety generated by blockages during the COVID-19 syndemic [50]. Therefore, developing this type of behavior could be associated with negative emotions because of anxiety and stress during confinement at home and the fear of gaining weight [51,52].

An interesting element in our results is that people reporting some form of ED had higher levels of restrictive eating behavior, external eating behavior, and eating compulsion during the syndemic when compared to the general population. In addition, they presented a higher frequency of purgatory eating behavior, a conduct almost non-existent in the general population. Results coincide with recent studies suggesting that the emergence of COVID-19 has impacted patients with ED, exacerbating their symptoms, even for subjects in remission [53]. These backgrounds would reinforce the idea that the consequences of the COVID-19 syndemic would highlight pre-existing symptoms of patients with a mental health disorder, and they would be more vulnerable to stress and anxiety produced by the syndemic [54,55].

The results reported in this review should be considered in the development of future research and public health policies helping to promote healthy habits and lifestyles to improve life quality and the well-being of people, especially during confinement at home; it is essential to minimize the negative consequences that restrictive physical contact measures are causing as a result of the COVID-19 syndemic [7,8,9,10,11]. In addition, it is essential to strengthen diagnostic and intervention programs for anxiety, stress, and depression, with special emphasis on patients who already have a diagnosis of mood or eating disorder.

To analyze the given evidence in this review, it is important to bear in mind the following considerations. The methodological design of all selected studies is cross-sectional, based on online self-reported questionnaires. The methodological quality of the studies was considered mostly with a moderate risk of bias, although there were two studies with a high risk of bias, and none of them was a randomized study. Furthermore, a significant heterogeneity of the data was found, which did not allow a meta-analysis. Although these antecedents represent a restriction in the generalization of the presented findings, there are other antecedents that compensate for these limitations; for example, subjects from different countries and continents participated, which would represent an adequate external validity of our findings. Furthermore, the total number of subjects who took part in the studies was over 20,000, which gives robustness to the sample. Finally, the syndemic context presented a great challenge in getting reliable assessments. Thus, the online self-reported questionnaire was the main tool used. In this sense, there is a growing body of evidence that validates self-assessment through the scales provided, offering adequate levels of reliability in measurements and forecasts [3,52,53,54].

## 5. Conclusions

The findings of this study suggest that people exposed to the preventive measures of restricting physical contact during the COVID-19 syndemic may experience changes in food intake. This would be shown in the consumption increase of foods considered healthy and unhealthy and in eating style alteration, which can vary between restrictive eating behaviors, uncontrollable eating, external feeding, and/or binging. These changes would be reflected in both healthy and unhealthy populations, however, they would be more highlighted in people who had a higher BMI, especially in those considered overweight and obese, and in people with an eating disorder. In addition, the increase in BMI during the syndemic may be a risk factor when choosing the type of diet, since there would be a tendency to prefer less healthy foods in subjects with a higher BMI.

## Figures and Tables

**Figure 1 nutrients-13-01168-f001:**
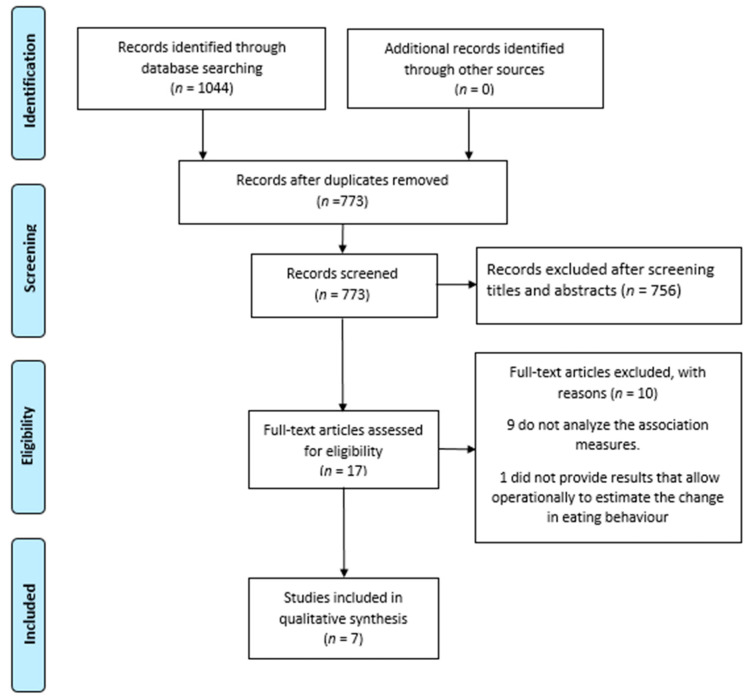
Article selection included in the analysis.

**Figure 2 nutrients-13-01168-f002:**
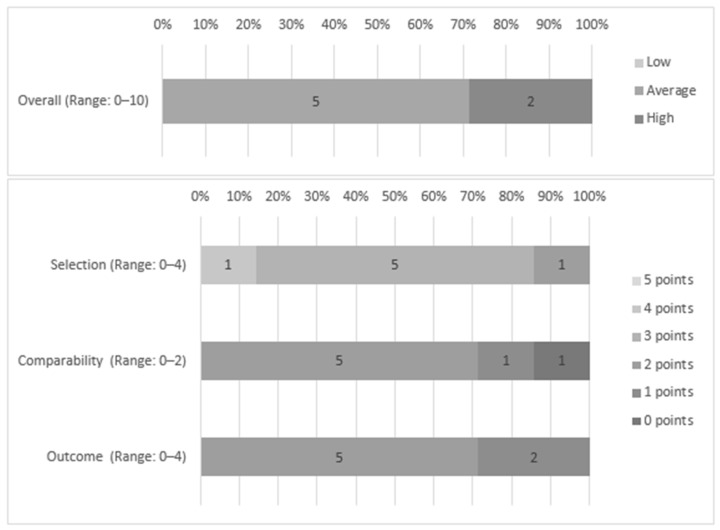
General and criteria scores of the studies through the Ottawa–Newcastle (NOS) scale.

**Table 1 nutrients-13-01168-t001:** Characteristics and primary results of cross-sectional studies.

Author(s), Year, and Country	Sample	Findings	
		Primary Outcome	Measure of Association
Ammar et al., 2020, international	1047 adults aged >18 years. Male 46%, female 54%. Asian 36%, African 40%, European 21%, and 3% other countries.	During home confinement, people eat more uncontrollably compared to before the pandemic. During confinement, food intake increased, especially snacks and unhealthy food, compared to before the pandemic.	*t* = −9.44, *p* < 0.001, d = 0.22 Increased number of main meals: *t* = −5.83, *p* < 0.001, d = 0.22. Snacks consumption increased: *t* = −6.89, *p* < 0.001, d = 0.30. Ingestion of unhealthy food increased: *t* = −3.46, *p* < 0.001, d = 0.14
Di Renzo et al., 2020, Italy	3553 subjects, with a mean age of 40.03 ± 13.53. Female 76.1%. Mean BMI 27.66 ± 4.1	During the COVID-19 emergency, there was a decrease in junk food intake; its intake was associated with a higher BMI and a younger age. During the COVID-19 emergency, the perception of weight gain was positive and inversely associated with a higher consumption of junk food and healthy food, a higher BMI, the female gender, and to a modified labor routine.	Decrease (29.8%) vs. increase (25,6%), *r*^2^ = 9.560, *p* = 0.002. BMI: OR = 1.025, *p* = 0.005. Age: OR = 0.979, *p* < 0.001 Junk food consumption: OR = 3.122, *p* < 0.001. Health food consumption: OR = 0.805, *p* = 0.002. BMI: OR = 1.073, *p* < 0.001. Modification of labor routine: OR = 1.250, *p* = 0.037
Phillipou et al., 2020, Australia	5289 adults, no report of ED, with a mean age of 40.62 ± 13.67. Female 80%. 180 adults with a self-report of ED, with a mean age of 30.47 ± 8.19. Female 95.6%. AN (*n* = 88), BN (*n* = 23), PBED (*n* = 6), EDNOS (*n* = 4), SE (*n* = 68).	Changes in eating styles were reported both in the population without ED and in those with ED, compared to the period before the COVID-19 emergency.	The levels of ERB and ECB increased 27.6% and 34.6% in people without ED, respectively, while the level of ERB, ECB, and FPB increased 64.5%, 35.5%, and 18.9% in people with ED, respectively.
Rodríguez-Pérez et al., 2020, Spain	7514 adults >18 years. Female 70.6%.	During the confinement period, there was an increase in the level of adherence to MedDiet, compared to the prior period, and an increase of 8% for every five days of greater confinement.	Prior confinement: Md = 6.0 (1–13). During confinement: Md = 7.0, (1–13), *p* < 0.001
Romeo-Arroyo, Mora & Vázquez-Araújo 2020, Spain	600 adults, with a mean age of 42.58 ± 12.25. Female 50.1%.	A change in the intake of different types of food was reported.	A little over 50% increase in the consumption of sweets; 30% to 40% enlarged consumption of meats, fruits, eggs, rice, pasta, bread, sausages and cold cuts, vegetables, and dairy products, and 33% decreased fish consumption.
Serin & Can Koç, 2020, Turkey	1064 adults >18 years. Female 58.64%.	The DEBQ results showed differences in “External Eating” behavior between people who were self-isolated during the pandemic versus those who were not.	Self-isolation Md = 26 (20–30) vs. no isolation Md = 23 (15–28.5), *p* < 0.01
Sidor and Rzymski, 2020, Poland	1097 adults with a mean age of 27.7 ± 9. Female 95.1%. Mean BMI of 23.5 ± 4.8	During quarantine, weight gain was observed, which was correlated with BMI and age, especially in overweight, obese, and individuals over 35. There was an increase in food intake that was associated with a higher BMI versus those with a lower BMI.	BMI: Rs = 0.21, *p* < 0.05 Age: Rs = 0.15, *p* < 0.05 BMI: M = 22.9 ± 4.6; M = 24.1 ± 5.1, *p* < 0.01

BMI = body mass index, ERB = eating restriction behavior, ECB = eating compulsive behavior, FPB = food purging behaviors, ED = eating disorder, AN = anorexia nervosa, BN = bulimia nervosa, PBED = periodic binge eating disorder, EDNOS = eating disorder not otherwise specified or other specified feeding, A = average, Md = median, NS = no specification of disorder, DEBQ = Dutch Eating Behavior Questionnaire, d = Cohen’s.

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
