# Peer review of "Consequences of the COVID-19 Syndemic for Nutritional Health: A Systematic Review"

_nutrients, 2021, doi:10.3390/nu13041168_

Round 1

Reviewer 1 Report

The objective of this Ms was to conduct a systematic review of the papers investigating the impact of preventive measures to restrict physical contact on nutrition health. The topic is clearly in the scope of Nutrients and the Ms is well written, pleasant to read and comprehensible. Nevertheless, I'm questioning the relevancy of the MeSH terms/inclusion criteria that have been used, as papers that I think contribute to this research topic have been omitted.

Abstract:

I suggest to mention the number of studies that were retained in the present review.

Introduction:

  • In addition to ref 15, to contextualize the topic of the review, the authors could also refer to the position paper from Rodgers R  et al. "The impact of the COVID-19 pandemic on eating disorder risk and symptoms" Int J Eat Disord 2020 who propose three pathways by which this pandemic may exacerbate unhealthy eating and thus eating disorders risk.
  • Please try to avoid paragraphs that correspond to a single sentence: I suggest to gather the 2 last paragraphs of the Introduction
  • I think there is one word missing in the sentence line 64, between " are not fully clarified, " and "is needed to ..."

Materials and Methods

  • Please provide the exact date when you stopped searching for articles in 2020: this could help to understand why some papers that could/should be mentioned in this review are not : were the following papers part of those identified (and were excluded)?: Flaudias V et al. "COVID-19 pandemic lockdown and problematic eating behaviors in a student population" J Behav Addictions 2020; Scharmer C et al. "Eating disorder pathology and compulsive exercise during the COVID-19 public health emergency- Examining risk associated with COVID-19 anxiety and intolerance of uncertainty" Int J Eat Disord 2020; Martinez-Ferran M et al. "Metabolic Impacts of Confinement during the COVID-19 Pandemic Due to Modified Diet and Physical Activity Habits" Nutrients 2020; Marty et al. "Food choice motives and the nutritional quality of diet during the COVID-19 lockdown in France" Appetite 2020; Puhl et al. "Weight Stigma as a Predictor of Distress and Maladaptive Eating Behaviors During COVID-19- Longitudinal Findings From the EAT Study" Ann Behav Med 2020; Rolland et al. "Global Changes and Factors of Increase in Caloric/Salty Food Intake, Screen Use, and Substance Use During the Early COVID-19 Containment Phase in the General Population in France: Survey Study" JMIR Public Health Surveill. 2020; Castellini et al. The impact of COVID-19 epidemic on eating disorders- A longitudinal observation of pre versus post psychopathological features in a sample of patients with eating disorders and a group of healthy controls Int J Eat Disord 2020; Panno et al. "COVID-19 Related Distress Is Associated With Alcohol Problems, Social Media and Food Addiction Symptoms- Insights From the Italian Experience During the Lockdown" Frontiers Psychiatry 2020.
  • My reading is that these papers could/should have been included, so another issue my be due to the MeSH/keywords that have been chosen: would these paper be part of those identified if using *eating* instead of eating behavior and eating habits?
  • it is not clear to me why papers on ED samples (eating disorders) were not included in the review

Results

  • Please note there is a typo error in Figure 1, in the box describing the articles excluded with reasons: should be "eating" and not "eiting"
  • Please note there are some typo errors in Table 1, study by Ammar et al. 2020, last column : replace "consumption" (written twice) by "consumption"; and "unhealth" by "unhealthy"
  • Please note there is one word missing in the sentence line 168: "One them" should be "One of them"

Discussion

If the paper I mentioned above as having being omitted from the review are not part of the review (and if the authors choose to exclude them please justify), the authors could at least refer to them in their Discussion. 

The authors mention the absence of a randomized study as a limit. As all the papers that were included concerned data collected during the COVID-19, I find it difficult to imagine how such a study design is possible. 

Please note there is one extra word line 241: "which in turn, it would influence" delete the "it".

Please note there is a word missing line 248: "people with BMI and older age" : isn't "higher" missing before "BMI"?

Author Response

Response to Reviewer 1 Comments

The authors are grateful for you taking time to assess the manuscript. We appreciate that. All concerns you raised are addressed below. All of us took time to address your suggestions. Some new ideas were added in the revised manuscript, which altered the number of lines, so the location  you made suggestions for correction may have been altered. However, you will find, in each of our answers, the number of the line where we did the requested corrections. Also, keep in mind that the alterations in the new manuscript were made with the Word tool "Alterations Control", so it will be easy for you to identify them. Cases in which this tool was not used, the changes were duly identified with red letters.

Point 1: I'm questioning the relevancy of the MeSH terms/inclusion criteria that have been used, as papers that I think contribute to this research topic have been omitted.

Response 1: Inclusion criteria and keywords, controlled by the MeSH vocabulary browser, were registered and validated by the Prospero International Database of Systematic Reviews (registration number is: CRD42020197618) to meet the objectives of our search.

Point 2: I suggest to mention the number of studies that were retained in the present review (Abstract).

Response 2: Lines 17, 18 has included the total number of studies selected for this review as requested by reviewer 1.

Point 3: In addition to ref 15, to contextualize the topic of the review, the authors could also refer to the position paper from Rodgers R  et al. "The impact of the COVID-19 pandemic on eating disorder risk and symptoms" Int J Eat Disord 2020 who propose three pathways by which this pandemic may exacerbate unhealthy eating and thus eating disorders risk.

Response 3: Suggested article is interesting and offers important information about the pandemic’s impact on feeding and eating disorders. However, the authors did not consider its use, since empirical studies were prioritized to justify the ideas planted by the reviewer.

Point 4: Please try to avoid paragraphs that correspond to a single sentence: I suggest to gather the 2 last paragraphs of the Introduction.

Response 4: Indicated paragraph (lines: 60-66) was rewritten, accepting the suggestion of reviewer 1 to gather the last two paragraphs.

Point 5: I think there is one word missing in the sentence line 64, between " are not fully clarified, " and "is needed to ..."

Response 5: Expression “is needed to understand its scope” was replaced by “understand its scope become necessary” to better comprehension (lines 61-63).

Point 6: Please provide the exact date when you stopped searching for articles in 2020: this could help to understand why some papers that could/should be mentioned in this review are not : were the following papers part of those identified (and were excluded)?: Flaudias V et al. "COVID-19 pandemic lockdown and problematic eating behaviors in a student population" J Behav Addictions 2020; Scharmer C et al. "Eating disorder pathology and compulsive exercise during the COVID-19 public health emergency- Examining risk associated with COVID-19 anxiety and intolerance of uncertainty" Int J Eat Disord 2020; Martinez-Ferran M et al. "Metabolic Impacts of Confinement during the COVID-19 Pandemic Due to Modified Diet and Physical Activity Habits" Nutrients 2020; Marty et al. "Food choice motives and the nutritional quality of diet during the COVID-19 lockdown in France" Appetite 2020; Puhl et al. "Weight Stigma as a Predictor of Distress and Maladaptive Eating Behaviors During COVID-19- Longitudinal Findings From the EAT Study" Ann Behav Med 2020; Rolland et al. "Global Changes and Factors of Increase in Caloric/Salty Food Intake, Screen Use, and Substance Use During the Early COVID-19 Containment Phase in the General Population in France: Survey Study" JMIR Public Health Surveill. 2020; Castellini et al. The impact of COVID-19 epidemic on eating disorders- A longitudinal observation of pre versus post psychopathological features in a sample of patients with eating disorders and a group of healthy controls Int J Eat Disord 2020; Panno et al. "COVID-19 Related Distress Is Associated With Alcohol Problems, Social Media and Food Addiction Symptoms- Insights From the Italian Experience During the Lockdown" Frontiers Psychiatry 2020.

My reading is that these papers could/should have been included, so another issue my be due to the MeSH/keywords that have been chosen: would these paper be part of those identified if using *eating* instead of eating behavior and eating habits?

it is not clear to me why papers on ED samples (eating disorders) were not included in the review.

Response 6: Exact starting and ending periods of the articles search to be included in this study were incorporated in line 80. This process considered studies published from January 2010 to July 2020. It should be noted, starting date of the search process registered in the Prospero International Review Database was 07/17/2020. Database accepted the revision proposal and delivered registration number CRD42020197618 on 07/23/2020. Thus, the date on which the reviewers stopped searching for studies in the electronic databases was 07/31/2020, one week after protocol be accepted. Regarding the completion of the review process, it was registered in the Prospero database on 10/26/2020.

Regarding the studies mentioned by reviewer 1, most of them did not enter because were published after the date on which researchers stopped searching in the electronic databases. Only one of the mentioned studies (Metabolic Impacts of Confinement during the COVID-19 Pandemic Due to Modified Diet and Physical Activity Habits) coincided with our search period. However, was not considered because did not meet the inclusion criteria of this review, since it is not an original study, but a review study.

Point 7: Please note there is a typo error in Figure 1, in the box describing the articles excluded with reasons: should be "eating" and not "eiting"

Response 7Typing error was corrected and figure 1, updated.

Point 8: Please note there are some typo errors in Table 1, study by Ammar et al. 2020, last column : replace "consumption" (written twice) by "consumption"; and "unhealth" by "unhealthy"

Response 8: Typographical errors were corrected as indicated by reviewer 1: One word “consumption” was replaced by “ingestion” and the word “unhealth” by “unhealthy”. Last result of this study was rewritten: "Ingestion of unhealthy food increased".

Point 9: Please note there is one word missing in the sentence line 168: "One them" should be "One of them"

Response 9: At line 184, typing error was corrected, according to the observation of reviewer 1.

Point 10: If the paper I mentioned above as having being omitted from the review are not part of the review (and if the authors choose to exclude them please justify), the authors could at least refer to them in their Discussion.

Response 10: Accepting suggestions of reviewer 1, following studies was incorporated, since the authors considered their results and conclusions contribute and enrich the discussion of this manuscript:

Line 267, Panno, A. et al., 2020.

Line 275, Puhl, R.M. et al., 2020.

Line 286, Flaudias, V. et al., 2020.

Line 296 Castellini, G. et al., 2020.

Point 11: Please note there is one extra word line 241: "which in turn, it would influence" delete the "it".

Response 11: Typing error in line 257 was corrected: word "it" removed.

Point 12: Please note there is a word missing line 248: "people with BMI and older age": isn't "higher" missing before "BMI"?

Response 12: At line 264, the text was rearranged to indicate the Mediterranean diet had less adherence as age and BMI increased.

Reviewer 2 Report

Brief summary:

The authors systematically reviewed the impact that preventive measures of physical contact restriction cause in the health nutrition. They concluded that nutrition health is affected by preventive measures to restrict physical contact as a result of the Covid-19 syndemic

The manuscript is well written and could be interesting for the readers and it is situated within the Journal Scope. I have some comments that could improve the final version of the manuscript. Before decision on the manuscript, the authors should give appropriate answer to comments at the below.

Broad comments

1. Although the topic is original and novel, the risk of bias of the reviewed studies is important.

2. Discussion: The authors should consider providing a more in-depth discussion of the results in the present study, and how this article addresses those results. Sometimes they just describe the results obtained by other authors, but they do not discuss their own results with them.

3. I recommend that the authors submit this manuscript to an editor for whom English is a first language. There are grammatical errors throughout the manuscript. While the large majority do not affect understanding, the article would be easier to read with corrections. Additionally, some phrasing is difficult to understand or changes the meaning such that it is incorrect as written.

Specific comments

- Search strategy: although the authors describe the inclusion criteria, I recommend to describe inclusion and exclusion criteria in a specific paragraph, detailing them. This change will improve de visibility for the readers

- Line 87-90: Authors should include the search equation developed, not only the keywords

- Authors should explain in a more in-depth way why did they include H1N1 and Ebola?

- Line 93: please indicate the author´s names

- Line 98: Was the concordance between the authors evaluate? For example, did they calculate the Kappa index? In the case of a discrepancy, what did they do?

- Data extraction: did the authors review the references of the included articles to search new potential articles to include?

- Line 103: please indicate the author´s names

- Methodological quality assessment: It would be useful to describe the risk of bias for each study individually, not just a general summary of all of them.

- Line 142: 7th instead of 7th

- Table 1: Authors specify that all the reviewed articles reported cross-sectional data so it is not necessary to include it in a column in the table. They could change the table title to “Characteristics and primary results of Cross-sectional studies”

- Line 218: “Eating Disorder (ED)” was previously defined

Author Response

Response to Reviewer 2 Comments

The authors are grateful for you taking time to assess the manuscript. We appreciate that. All concerns you raised are addressed below. All of us took time to address your suggestions. Some new ideas were added in the revised manuscript, which altered the number of lines, so the location you made suggestions for correction may have been altered. However, you will find, in each of our answers, the number of the line where we did the requested corrections. Also, keep in mind that the alterations in the new manuscript were made with the Word tool "Alterations Control", so it will be easy for you to identify them. Cases in which this tool was not used, the changes were duly identified with red letters.

Point 1: The authors should consider providing a more in-depth discussion of the results in the present study, and how this article addresses those results. Sometimes they just describe the results obtained by other authors, but they do not discuss their own results with them 

Response 1: Accepting suggestions of reviewer 2, new studies was incorporated into the manuscript, contributing and enriching discussion. New references were incorporated into the following lines:

Line 267, Panno, A. et al., 2020.

Line 275, Puhl, R.M. et al., 2020.

Line 286, Flaudias, V. et al., 2020.

Line 296 Castellini, G. et al., 2020.

Point 2: Search strategy: although the authors describe the inclusion criteria, I recommend to describe inclusion and exclusion criteria in a specific paragraph, detailing them. This change will improve de visibility for the readers

 Response 2: Following recommendations of reviewer 2, a separate paragraph (lines 85-90) was included, and exclusion criteria are detailed.

Point 3: Line 87-90, authors should include the search equation developed, not only the keywords.

Response 3: Lines 98 and 99 detail searching strategy. Additionally, appendix A, Table A1, was attached at line 342 for better understanding of this strategy.

Point 4: Authors should explain in a more in-depth way why did they include H1N1 and Ebola?

Response 4: Following recommendations of reviewer 2, this paragraph was rewritten, including an explanation about the use of these two concepts as keywords (lines 91-97).

These words were included as a way of doing a broader and retrospective search, since these two viruses have some similar characteristics to SARS-CoV-2: in the transmission routes and in the impact on the population.

Point 5: Line 93: please indicate the author’s names

Response 5: At lines 102, 107 and 110, reviewers initials who performed a preliminary reading were included, as requested by reviewer 2.

Point 6: Line 98: Was the concordance between the authors evaluate? For example, did they calculate the Kappa index? In the case of a discrepancy, what did they do?

Response 6: In lines 108-110 is explained how discrepancies between the 2 reviewers were solved. Basically, each reviewer independently carried out studies selection. When were no concordance in the study selection, a third reviewer performed the judge's work, trying to seek consensus between the two reviewers. In cases of no consensus, the 3rd judge, considering the arguments of the two initial reviewers, finally decided to include the study or not.

Point 7: Data extraction: did the authors review the references of the included articles to search new potential articles to include?

Response 7: Since Cochrane Group's Methodological Guide for RAPID REVIEWS was followed, manual literature review was not carried out in this case. In protocol, originally approved by Prospero International Database of Systematic Reviews (under registration number CRD42020197618), a period of three months was considered to accomplish this review, considering the urgency of having information that allows better understanding the scope of this syndemic.

Point 8: Line 103: please indicate the author´s names

Response 8: At line 113, initials of reviewers who performed data extraction were included, as requested by reviewer 2.

Point 9: Methodological quality assessment: It would be useful to describe the risk osbias for each study individually, not just a general summary of all of them.

Response 9: The paragraph was rewritten (lines 140-148) and a detailed description of the risk of bias was made, considering criteria of the NOS scale. In addition, line 334 was attached, in appendix B (Figure B1), with detailed scores for each item in NOS scale.

Point 10: Line 142: 7th instead of 7th

Response 10: Correction made on line 158: “7th” was replaced by “one”.

Point 11: Authors specify that all the reviewed articles reported cross-sectional data so it is not necessary to include it in a column in the table. They could change the table title to “Characteristics and primary results of Cross-sectional studies”

Response 11: The suggestion of reviewer 2 was accepted. The second column of table 1 was eliminated. At line 178, title was changed.

Point 12: Line 218: “Eating Disorder (ED)” was previously defined

Response 12: Line 234 was corrected, as indicated by reviewer 2.

Round 2

Reviewer 1 Report

Thank you for your responses and the changes that have been made. Please note that you may have uploaded the old version of the figure : the typo error is still there (eiting instead of eating).

Author Response

Dear reviewer 2, first of all, I would like to thank you once again for your time and willingness in evaluating our manuscript.
Second, to inform that the unfortunate error in Figure 1 has already been corrected, which was updated in the new manuscript uploaded today to the MDPI platform.

Reviewer 2 Report

The authors´ answers and changes were globally satisfactory. There is only one point to clarify.

- Figure 1 and 2 are duplicated in the manuscript and in the supplementary files. Please correct it

Author Response

Dear reviewer, first, I would like to thank you once more for your time and disposition in evaluating our manuscript.
Second, to report that unfortunate duplicated information error in Figures 1 and 2 has already been corrected. Therefore, following your instructions, only the information corresponding to Appendix A and B remained in the supplementary material.